# Simple and Fast Pesticide Nanosensors: Example of Surface Plasmon Resonance Coumaphos Nanosensor

**DOI:** 10.3390/mi14040707

**Published:** 2023-03-23

**Authors:** Beste Oymen, Mitra Jalilzadeh, Fatma Yılmaz, Süleyman Aşır, Deniz Türkmen, Adil Denizli

**Affiliations:** 1Department of Gastronomy and Culinary Arts, Bahçeşehir Cyprus University, Mersin 10 Turkey, Nicosia 99010, Cyprus; 2Department of Chemistry, Faculty of Science, Hacettepe University, Beytepe, Ankara 06800, Turkey; 3Chemistry Technology Division, Vocational School of Gerede, Bolu Abant Izzet Baysal University, Bolu 14030, Turkey; 4Department of Materials Science and Nanotechnology Engineering, Near East University, Mersin 10 Turkey, Nicosia 99138, Cyprus

**Keywords:** molecular imprinting, surface plasmon resonance, nanosensor, nanofilm, coumaphos

## Abstract

Here, a molecular imprinting technique was employed to create an SPR-based nanosensor for the selective and sensitive detection of organophosphate-based coumaphos, a toxic insecticide/veterinary drug often used. To achieve this, UV polymerization was used to create polymeric nanofilms using *N*-methacryloyl-l-cysteine methyl ester, ethylene glycol dimethacrylate, and 2-hydroxyethyl methacrylate, which are functional monomers, cross-linkers, and hydrophilicity enabling agents, respectively. Several methods, including scanning electron microscopy (SEM), atomic force microscopy (AFM), and contact angle (CA) analyses, were used to characterize the nanofilms. Using coumaphos-imprinted SPR (CIP-SPR) and non-imprinted SPR (NIP-SPR) nanosensor chips, the kinetic evaluations of coumaphos sensing were investigated. The created CIP-SPR nanosensor demonstrated high selectivity to the coumaphos molecule compared to similar competitor molecules, including diazinon, pirimiphos-methyl, pyridaphenthion, phosalone, *N*-2,4(dimethylphenyl) formamide, 2,4-dimethylaniline, dimethoate, and phosmet. Additionally, there is a magnificent linear relationship for the concentration range of 0.1–250 ppb, with a low limit of detection (LOD) and limit of quantification (LOQ) of 0.001 and 0.003 ppb, respectively, and a high imprinting factor (I.F.4.4) for coumaphos. The Langmuir adsorption model is the best appropriate thermodynamic approach for the nanosensor. Intraday trials were performed three times with five repetitions to statistically evaluate the CIP-SPR nanosensor’s reusability. Reusability investigations for the two weeks of interday analyses also indicated the three-dimensional stability of the CIP-SPR nanosensor. The remarkable reusability and reproducibility of the procedure are indicated by an RSD% result of less than 1.5. Therefore, it has been determined that the generated CIP-SPR nanosensors are highly selective, rapidly responsive, simple to use, reusable, and sensitive for coumaphos detection in an aqueous solution. An amino acid, which was used to detect coumaphos, included a CIP-SPR nanosensor manufactured without complicated coupling methods and labelling processes. Liquid chromatography with tandem mass spectrometry (LC/MS-MS) studies was performed for the validation studies of the SPR.

## 1. Introduction

Pesticides are well-known pollutants in the food industry [1,2,3]. In agriculture, aquaculture, apiculture and animal husbandry, pesticides have been used to kill or eliminate a type of “pest” to increase the production and shelf life of foods, according to target pest type pesticides classified as herbicides, fungicides, rodenticides, acaricides, molluscicides, pediculicides, insecticides, etc. [4]. Recent reports show that more than three billion kilograms of pesticides are sold and used globally, and their usage is still rising [4]. Coumaphos is an organophosphorus-based insecticide. It is used in veterinary practices in animal husbandry to reduce ticks, mites, and lice, as they may carry diseases and lower production. It has been used against *Varroa jocobsoni* or *Varroa destructor* infestation in apiculture. Varroa mites are among the most destructive ectoparasites for honey bee hives [5]. The mite’s in an infested pack can harm bee colonies directly by feeding on the fat bodies (type of storage insect tissue) of adult, pupal and larval bees, causing tissue damage [6] and indirectly by transmitting viruses such as Deformed Wing Virus and Kashmir bee virus [7,8]. Additionally, it is noted that if the infestation is left untreated, it usually results in the hive’s demise.

Coumaphos is an organophosphorus compound used as an insecticide frequently by beekeepers to manage parasitic mites [9]. Even though pesticides such as coumaphos are used to protect bee colonies against mite infestation, it poses a risk for bees and humans [10,11]. It is reported that coumaphos usage results in contamination in apicultural products; wax, honey, pollen and their processed products; ointments, candies and cream lotions [12,13,14]. According to the European Union (EU), a risk assignment and regulation maximum residue limit for coumaphos in the apicultural product is proposed as 0.1 mg/kg [15,16]. For a 70 kg human, the fatal oral dosage ranges from 50 to 500 mg/kg (US Environmental Protection Agency (EPA) 1998, Extremely Hazardous Substances (EHS), Chemical Profiles and Emergency First Aid Guides. Washington, DC, US, Government Printing Office) [17]. In humans, many reports of acute poisoning of organophosphates research suggested that organophosphate acts similar to nerve agents and may cause muscle failure, resulting in a patient’s death [18,19,20].

Samples are analysed periodically to control contaminants such as coumaphos to evaluate and reduce risk in the food market. Frequently in quality control laboratories, classical instrumental techniques are employed, including gas chromatography (GC) and liquid chromatography (LC) connected with conventional or mass spectrometer detectors [21,22]. Although these methods are reliable and commonly available, they require costly equipment, experienced personnel, and high maintenance fee. Additionally, their analysis takes a long time and requires pre-processing because of the complicated analysis matrix that foods possess. Additionally, it was reported that the elapsed time between food production and the shipping conditions could cause food contamination with fungi and other microorganisms, resulting in unreliable analysis results of the quality control laboratory [23]. These drawbacks, with trends in foodstuff production rising along with the human population, created an urgent demand for new precise and fast analysis techniques to maintain food quality and safety.

Molecularly imprinted polymers (MIPs) are synthetic polymers that mimic the behaviour of natural receptor binding sites by having specific recognition sites for the target analyte. Imprinted polymers used in the selective detection of target molecules are customizable materials and have applications in chemical separations, drug delivery, sensors and catalysis systems [24]. As an alternative artificial receptor, molecularly imprinted polymer-based sensors have been utilized recently. Traditional detection technologies have the disadvantages of being time-consuming and requiring extensive and costly equipment. They are not suitable for consumer use, so the production of devices that allow rapid, reliable and inexpensive detection of pesticides is essential.

In recent years, sensor-based technologies have responded to that demand and presented new detection methods for contaminant detection. Because of their ease of use, high specificity and sensitivity, real-time detection, low price, and lack of labelling, surface plasmon resonance (SPR) sensors have been used to detect several biomolecules. MIPs, which are simple to prepare, affordable, stable, and capable of recognizing molecules, are also utilized to create specific binding sites for target molecules on SPR sensors.

Surface plasmon resonance (SPR)-based sensors are popular for residue detection [25,26]. SPR sensors use an optical phenomenon called plasmonic resonance to detect changes on the chip surface. Binding events on the surface change intensity shift on the reflectance, and this change could be detected even for trace binding amounts.

The main factor for the sensor’s sensitivity is based on the effectiveness of the receptors. Additionally, the non-destructive nature and detection capability made SPR sensors a good choice for sensors. Using biological receptors, enzymes, proteins, lectins, etc., offers the high affinity desired while their drawbacks include fragility, price, hard-to-supply and limited analyte type to be employed. These factors limit the usage of biological receptors in the field of sensors.

Studies on biomolecular interactions widely use MIPs-based sensor systems for biological applications. Combining MIPs with the high binding ability and stability against pH, temperature, pressure and ion concentration changes with SPR made them popular for sensor development. Additionally, the low fabrication price of MIPs and ease of synthesis made sensors based on MIPs a good choice for detecting contaminants [27,28]. Additionally, MIP-based sensors provide opportunities for efficient, sensitive and affordable sensing methods using innovative miniaturized equipment, overcoming the various limitations of traditional sensing techniques.

We analysed 100 honey samples in our previous research and obtained exciting results [29]. According to the LC/MS-MS studies, coumaphos residues were the highest (%29) detected pesticides in the honey samples. Therefore, in this work, we created a very selective, quick, and precise analysis approach to identify the commonly used coumaphos pesticide in honey samples. The sensitivity of the CIP-SPR nanosensor was increased by combining molecular imprinting technology with the plasmonic characteristics of SPR. Herein, *N*-methacryloyl-l-cysteine methyl ester (MAC) coordinating with coumaphos pesticide was used as a functional monomer to obtain a CIP-SPR nanosensor. Scanning electron microscopy (SEM), atomic force microscopy (AFM) and contact angle (CA) measurements were used to characterize the CIP-SPR and NIP-SPR nanosensors. The selectivity of the CIP-SPR nanosensor was studied using competing molecules, diazinon, pirimiphos-methyl, pyridaphenthion, phosalone, *N*-(2,4-dimethylphenyl)formamide, 2,4-dimethylaniline, dimethoate, phosmet, amitraz and parathion-ethyl. Using the non-imprinted NIP-SPR nanosensor for the comparison studies, the imprinting efficiency for the CIP-SPR nanosensor was examined. Coumaphos solutions were passed through a CIP-SPR nanosensor with a concentration range of 0.1 to 250 ppb to assess the adsorption kinetics. Reusability tests were reported for the 100 ppb coumaphos solutions, which were used six times consecutively. Additionally, real sample analyses were performed to evaluate the impacts of existing residues.

## 2. Materials and Method

### 2.1. Materials and Instruments

Standards of coumaphos, diazinon, pirimiphos-methyl, pyridaphenthion, phosalone, *N*-(2,4-dimethyl phenyl)formamide, 2,4-dimethylaniline, dimethoate, phosmet, amitraz and parathion- ethyl with 94–99% purity were provided from Dr Ehrenstorfer Co. (Milton, ON, Canada). Each pesticide’s 1000 µg/mL stock solution was prepared by dissolving the respective standard in methanol or acetonitrile and storing the resulting mixture in amber glassware at a temperature and humidity- and light-protected storage at −18 °C. 2-Propene-1-thiol, 2-hydroxyethyl methacrylate, ethylene glycol dimethacrylate, and azobisisobutyronitrile were purchased from Sigma Chemical Co. (St. Louis, MO, USA). *N*-methacryloyl-l-cysteine methyl ester (MAC) was supplied from Nanoreg (Ankara, Turkey). The SPR chips were obtained from GWC Tech (Madison, WI, USA).

The mole ratio of functional MAC monomer and coumaphos molecule was determined by UV-VIS spectrophotometer (SHIMADZU UV-1601, Tokyo, Japan). Characterization of the CIP-SPR and NIP-SPR nanosensors were performed by SEM (Jeol JEM 1200 EX, Tokyo, Japan), AFM (Oxford, UK), and CA (KRUSS DSA100, Hamburg, Germany) measurements. A SPR imager II (GWC Technologies, Madison, WI, USA) monitoring the angle shifts of incident light at the surface resonance caused by a CIP-SPR nanosensor was used for kinetic experiments.

### 2.2. Preparation of CIP-SPR and NIP-SPR Nanosensors for the Detection of Coumaphos

Coumaphos-methacryloyl-l-cysteine methyl ester (MAC) pre-complex was synthesized before designing a CIP-SPR nanosensor. According to this purpose, to obtain the stoichiometric ratio of this complex, the coumaphos–MAC complex was prepared in 1:0.5, 1:1, 1:2, and 1:4 molar ratios. The best stoichiometric ratio of this complex was tested by the absorbance of the coumaphos–MAC complex measured in the range of 200–700 nm wavelength by spectrophotometer. The maximum absorbance value of the coumaphos–MAC complex was observed in a 1:1 stoichiometric ratio.

The cleaning process of SPR chips was first performed with a piranha solution (H_2_SO_4_:H_2_O_2_, 3*v*/1*v*) and a few drops of ethyl alcohol in deionized water. At last, these chips were dried in a vacuum oven (200 mmHg, 40 °C) for 2 h. To add allyl groups on the gold surface of the SPR chip, allyl mercaptan (≥90) was dropped and incubated for 12 h. The removal of physically bound allyl mercaptan molecules was performed by a deionized water and alcohol mixture. After drying in a vacuum oven (200 mmHg, 30 °C), coumaphos imprinted and non-imprinted nanofilm was modified on the SPR chips’ surfaces. To prepare the CIP-SPR, a stock solution of monomer was prepared in a ratio of 12.6% HEMA, 18.1% EGDMA, and 69.3% of coumaphos-MAC complex (1:1, Molar ratio). After adding 4 mg of AIBN, 3 µL of this solution was dropped onto the SPR chip that formerly added allyl groups. To initiate the polymerization, the chip was taken under UV light (100 W, 365 nm) for 45 min at 25 °C. The chip surface was cleaned with deionized water and ethyl alcohol after the polymerization process, dried in a vacuum oven, and then kept in a desiccator until usage. The exact process was applied to prepare the NIP-SPR nanosensor without adding coumaphos. To remove coumaphos from the CIP-SPR nanosensor, NaOH solution (0.5 M) was used. A NIP-SPR nanosensor was prepared using the same recipe without adding a coumaphos molecule.

### 2.3. Characterization of Designed CIP-SPR Nanosensor Chips

CIP-SPR nanosensors were characterized with SEM, AFM, and contact angle measurement methods. SEM images were recorded with 10.00 kx and 20.00 kx magnification ratios. An ambient AFM instrument (Nanomagnetic instrument, Axford, UK) was applied for AFM Studies. AFM images were taken in dynamic mode at 2 µm/s scanning rate and 256 × 256 pixels resolution, and the designed SPR chips were also attached to the sample holder by a double-sided carbon strip. A sessile drop system was used for the CA measurements.

### 2.4. Applying CIP-SPR Nanosensor for Detection of Coumaphos and Kinetic Studies with Designed CIP-SPR Nanosensor

Nanosensors were washed with deionized water after characterising the designed CIP-SPR and NIP-SPR nanosensor chips. Then, imprinted coumaphos was desorbed from the CIP-SPR nanosensor chip by applying the 0.5 M NaOH solution. The detection experiments were performed at a 120 µL/min flow rate throughout the analysis. The first detection experiments were performed in the different medium pHs (5.0, 6.0, 7.4, 8.0) with the same concentration of coumaphos (100 ppb) to determine the optimum medium pH recorded as 7.4. After applying coumaphos to the SPR system for 0.1–250 ppb concentration range in the 7.4 pH medium, the pH 7.4 phosphate buffer was used equilibrating of designed imprinted and non-imprinted SPR chips, and 0.5 M NaOH solution was used for the desorption of nanosensor chips. The selectivity of the CIP-SPR nanosensor was tested by applying diazinon, pirimiphos-methyl, pyridaphenthion, phosalone, *N*-2,4(dimethylphenyl)formamide, 2,4- dimethylaniline, dimethoate, and phosmet solutions to the CIP-SPR nanosensor. The reusability and repeatability of the CIP-SPR nanosensor were examined by six times equilibration–adsorption–desorption cycles using an aqueous solution of coumaphos (100 ppb). Additionally, reusability studies were examined for different times intervals (first, second, seventh, and fourteenth days). In addition, analyses were carried out with real samples to investigate the effects of existing residues. Because the presence of the residues in the analysed matrix effects the signal response, all honey stocks were diluted with deionized water in a 1:5 ratio. The matrices used in this study were honey collected from organically grown farms purchased from a local store.

### 2.5. Validation Studies

Validation studies were conducted according to our previous study [29] and briefly explained below. The liquid chromatography system of Shimadzu equipped with an autosampler (Nexera SIL-20ACXR), a column oven (Shimadzu CTO-10ASvp) and a double pump (Nexera XR, LC-20ADXR) was handled (Kyoto, Japan). A SynergiTM 2.5 µm Fusion-RP 100 Å 50 × 2 mm column (Phenomenex, Torrance, NJ, USA) was used for the chromatographic isolation. The mobile phase A (5.0 mM ammonium formate in water) and phase B (5.0 mM ammonium formate in methanol) were utilized as mobile phases, and the flow rate was 400 µL/min. Additionally, 40 °C was the temperature of the column. The gradient elution program launched with a 95% water mobile phase, dropped to 5% in 6.5 min and was retained for 1.5 min, then rose to 95% and was retained until the analysis finished. The duration for LC analysis was 12 min., and 5 µL was the injection volume.

Mass spectrometrometric analysis was implemented by a Shimadzu 8060 Triple Quadruple Mass Spectrometer (Kyoto, Japan) performed in MS/MS mode operated with multiple reaction monitoring (MRM). Electrospray ionization (ESI) with ion modes negative and positive was used. The software Lab Solutions (Version 5.86 SPI) tested operation and data acquisition as an instrument setting. The ion source parameters were retained at 300 °C was an interface temperature, 10 L/min was a flow heating gas, 3 L/min was a nebulizing gas flow, 10 L/min was a drying gas flow, 3 kV was an interface voltage, 250 °C was a desolvation line (DL) temperature, 270 kPa was a collision-induced dissociation (CID) gas, 2.06 kV was a detector voltage, and 400 °C was the heat block temperature. The matrix-matched calibration curve was employed to calculate each pesticide’s amount. Two MRM transitions were used for each analyte, one for quantification and the other for qualification to prevent false detection.

## 3. Results and Discussion

### 3.1. Characterization of CIP-SPR and NIP-SPR Nanosensor

Nanofilm formation was performed via bulk polymerization of EGDMA, HEMA, and a coumaphos–MAC precomplex onto the nanosensor chip surface. SEM, AFM, and contact angle measurements were examined to characterise the CIP-SPR and NIP-SPR nanosensor chips.

The surface hydrophilicity of the CIP-SPR and NIP-SPR nanosensors were evaluated by measuring contact angle values using the sessile drop system. Ten different areas of the nanosensor surfaces were used to estimate the water contact angle values for the CIP-SPR nanosensors, and average CAs were calculated. CA values were estimated as 63.4° ± 0.2 and 37.3° ± 0.5 for the surface of CIP-SPR and NIP-SPR nanosensors, respectively. When the CA images of the CIP-SPR nanosensor (Figure 1A) and NIP-SPR nanosensor (Figure 1B) were evaluated, it was deduced that CA increased as the hydrophobicity of the surface increased as a result of coumaphos incorporation into the nanofilm structure because of molecular imprinting.

The surface depths of the CIP-SPR nanosensor were investigated using AFM in tapping mode. AFM images showing the three-dimensional surface structure of the bare SPR chip surface, ally-modified SPR chip surface, and coumaphos-imprinted CIP-SPR nanosensor chip surface are shown in Figure 2A–C, respectively. The AFM images determined the surface depth of the bare SPR chip, ally-modified SPR chip, and CIP-SPR nanosensor as 9.68 ± 0.96 nm, 16.70 ± 2.17 nm and 64.30 ± 1.71 nm, respectively. The difference between the surface depth values of the bare SPR chip and the CIP-SPR nanosensor indicates that the imprinted polymeric nanostructure was incorporated onto the chip surface successfully.

SEM micrographs of the CIP-SPR nanosensor were reported with 10.00 kx (Figure 3A) and 20.00 kx (Figure 3B) magnification ratios. From the images, it was deduced that the molecular imprinting process had changed the size of the cavities of the CIP-SPR nanosensor, which was used as a plasmonic structure-based chemical nanosensor.

### 3.2. Kinetic Studies

Absorbance-, fluorescence-, and chemiluminescence-based optical nanosensors are effective analytical and detection tools. Adsorption kinetic analyses of CIP-SPR and NIP-SPR nanosensors were performed by passing coumaphos solutions through the CIP-SPR and NIP-SPR nanosensors with a flow rate of 150 μL/min and an operating wavelength of 800 nm. SPR imager II software system was used for data analyses of Coumaphos.

The effect of pH in detecting coumaphos by the CIP-SPR nanosensor was examined by performing kinetic analysis in four different pH values (4.0, 6.0, 7.4, 8.0) of phosphate buffer, according to Figure 4. The optimum pH value for coumaphos detection by the CIP-SPR nanosensor was determined as pH 7.4. As expected, the complexation of coumaphos and functional monomer is affected by the medium pH. Because medium pH causes the protonation and deprotonation of the functional monomer. The increase of the sensorgram signal with increasing medium pH is related to the deprotonation of the functional monomer and complexation of coumaphos. The decrease of the sensorgram signal in the basic medium is because of the association decrement of coumaphos due to the deprotonation of the MAC functional monomer in the surface of the SPR chip.

The real-time detection experiments of coumaphos molecules by designed CIP-SPR nanosensors were performed in a pH 7.4 phosphate buffer solution. After equilibrating the designed CIP-SPR nanosensor with a pH 7.4 phosphate buffer solution at the 120 µL/min flow rate, these experiments were performed in different concentrations of coumaphos (0.1–250 ppb), and sensorgrams are shown in Figure 5A. A standard calibration curve for coumaphos detection by the designed CIP-SPR nanosensor in different concentrations is given in Figure 5B.

The calculated limit of detection (LOD: 0.001) and limit of quantification (LOQ: 0.003) values were determined by using the standard deviation of the designed CIP-SPR nanosensor response.

A pseudo-first-order kinetic model can define the adsorption of coumaphos by the designed CIP-SPR nanosensor, the rate of analyte assumed constant in the flow cell (Equation (1)).
dΔR/dt = k_a_C·ΔR_max_ − (k_a_C + k_d_)·ΔR(1)

That ΔR is the signal response changing of the SPR chip, and ΔR_max_ is the maximum amount of signal response measured by the instrument. C (ppb) is the concentration of coumaphos, k_a_ is the association rate constant, and k_d_ is the dissociation rate constant.

To obtain k_a_ and k_d_ values, primarily a plot of changing rate of the signal response (ΔR/dt) versus ΔR was used to obtain S values, and then a plot of S versus C was used as follows:S = k_a_C + k_d_(2)

The association and dissociation constant (K_a_, K_d_) can be calculated as K_a_ = k_a_/k_d_, also Scatchard equation obtained by an equilibrium of Equation (1) dΔR/dt = 0, the equation can be rewritten as:ΔR_eq/_C = K_a_ΔR_max_ − K_a_ΔR_eq_(3)

Kinetic parameters of real-time detection experiments are given in Table 1.

Table 1 shows the association rate constant (k_a_: 0.0013 L/µgs) and dissociation rate constant (k_d_: 5 × 10^−4^ 1/s) of interaction between coumaphos and the CIP-SPR nanosensor, which demonstrate a tendency to association between the coumaphos molecules and CIP-SPR nanosensor. In addition, the equilibrium association constant (K_a_: 2.6 L/µg) and the dissociation constant (K_d_: 0.38 µg/L) represent the high binding affinity between the analyte and nanosensor.

### 3.3. Isotherms of Real-Time Detection

The interaction between coumaphos molecules and coumaphos-imprinted nanofilm, and also the adsorption behaviour of these molecules were thermodynamically examined by three different isotherm models:
ΔR_eq/_C = K_a_ΔR_max_ − K_a_ΔR_eq_        Langmuir(4)

ΔR = ΔR_max_·C^1/n^             Freundlich(5)
ΔR = {(ΔR_max_·C^1/n^/K_d_) + C^1/n^}  Langmuir–Freundlich(6)

The heterogeneity index of the Freundlich isotherm is referred to as 1/n. Additionally, the ΔR is the changing of the SPR chip response signal. In addition, K_d_ and K_a_ are association and dissociation constants, respectively.

The Langmuir and Freundlich adsorption models explain the homogeneous monolayer adsorption and heterogeneous multilayer binding sites, respectively. The most imprinted binding sites are fitted to the Freundlich model, meaning heterogeneous binding sites are present dominantly. The Freundlich adsorption isotherm model best fits the imprinted binding sites for the low concentration range. Still, the Langmuir–Freundlich isotherm model can explain the concentration region’s imprinted binding site behaviour.

The Langmuir, Freundlich, and Langmuir–Freundlich models isotherm models were applied to the experimental data. As seen in Table 2, the correlation coefficient of all isotherms is high (≥0.97). The behaviour of binding sites for the adsorption of coumaphos to the coumaphos imprinted nanofilm is best fitted to the Langmuir model isotherm, which means that this adsorption is monolayer but a non-linear Scatchard plot displaying the existence of heterogeneous binding sites. This heterogeneity can be explained by the difference in binding sites on the surface of the chip that has a similar affinity to coumaphos molecules. The correlation coefficients of Freundlich and Freundlich–Langmuir model isotherms are high, but not higher than the Langmuir model, which exhibits that the designed CIP-SPR nanosensor has some heterogeneous binding sites with the same affinity to coumaphos molecules, and also the high binding capacity of the CIP-SPR nanosensor to the coumaphos molecules can explain the behaviour of the different concentration ranges applied to the imprinted binding sites. Additionally, the value of heterogeneity (1/n < 1) supports the fitting of the Langmuir model to this adsorption.

### 3.4. Selectivity Studies

The specificity of the CIP-SPR nanosensor for coumaphos detection was examined by applying coumaphos-imprinted and non-imprinted SPR nanosensors to the other competitor molecules solution (50 ppb) as diazinon, pirimiphos-methyl, pyridaphenthion, phosalone, *N*-(2,4-dimethylphenyl)formamide, 2,4-dimethylaniline, dimethoate, phosmet, parathion ethyl, and amitraz. According to the experimental results (Table 3 and Figure 6), the adsorption of coumaphos molecules to the CIP-SPR nanosensor is higher than the other competitor molecules and the NIP-SPR nanosensor. The selectivity coefficients (k) and relative selectivity coefficients (k′) for coumaphos pesticide were recorded individually by applying the competitor molecules (diazinon, pirimiphos-methyl, pyridaphenthion, phosalone, *N*-(2,4-dimethylphenyl)formamide, 2,4-dimethylaniline, dimethoate, phosmet, amitraz, and parathion-ethyl) to the CIP-SPR and NIP-SPR nanosensors, respectively. As seen in Table 3, the selectivity constants calculated as k = ΔRcoumaphos/ΔRcompetitor for the CIP-SPR nanosensor are higher than the NIP-SPR nanosensor, and these results demonstrate that the designed CIP-SPR nanosensor is more selective to the coumaphos molecules. As seen in Table 3, the selectivity coefficients for coumaphos are higher for the CIP-SPR nanosensor than for the NIP-SPR nanosensor. This suggests that the CIP-SPR nanosensor has a higher affinity towards coumaphos compared to the non-imprinted nanosensor. The relative selectivity coefficients (k′) for coumaphos are also higher for the CIP-SPR nanosensor than the NIP-SPR nanosensor. This indicates that the CIP-SPR nanosensor is more selective towards coumaphos than the other competing molecules. Overall, the table shows that the CIP-SPR nanosensor is more selective for coumaphos and has the potential to be used as a sensitive and selective method for coumaphos detection in various environmental mediums and food samples.

As seen in Table 3, CIP-SPR nanosensor 7.1, 3.8, 5.4, 3.9, 5.3, 8.3, 2.9, 23.07, 24.6, and 6.5 times more selective for the template coumaphos pesticide than competitor Diazinon, Pirimiphos-methyl Pyridaphenthion, Phosalone, *N*-(2,4-dimethylphenyl)formamide, 2,4-dimethylaniline, Dimethoate, Phosmet, Amitraz and Parathion-ethyl molecules. The high ratio of selectivity values indicates that the CIP-SPR nanosensor was successfully prepared for selective coumaphos pesticide detection. The imprinting efficiency of the CIP SPR nanosensor for coumaphos detection was evaluated by comparing it with the non-imprinted NIP SPR nanosensor to indicate the specificity of the prepared CIP SPR nanosensor for coumaphos detection. The high value of the calculated imprinting factor (I.F.4.4) proved that the CIP SPR nanosensor detects coumaphos pesticide specifically. Additionally, the linear relationship for the 0.1–250 ppb coumaphos concentration range reaching low limits of detection and limit of quantification (LOQ: 0.001 and LOQ: 0.003 ppb) values indicates that the MIP combined with the SPR method was used successfully for the specific determination of coumaphos pesticide. Overall, the table shows that the CIP-SPR nanosensor is more selective for coumaphos and has the potential to be used as a sensitive and selective method for coumaphos detection in various environmental mediums and food samples.

### 3.5. Reusability and Repeatability

The real-time detection of coumaphos by the CIP-SPR nanosensor was performed at four different times in two weeks for the same concentration to indicate the reusability and stability of the CIP-SPR nanosensor after long-term storage conditions. As seen in Figure 7A, the activity of the designed CIP-SPR nanosensor after two weeks is 87% of the first-day adsorption capacity of the designed CIP-SPR nanosensor. Additionally, the repeatability of the designed CIP-SPR nanosensor was examined by a six-time repetition of the equilibration–adsorption–desorption cycle for the detection of coumaphos by CIP-SPR nanosensor. As seen in Figure 7B, the designed CIP-SPR can be used repeatedly without the loss of binding and performance.

### 3.6. Validation Studies

In LC/MS-MS, recovery studies of coumaphos were performed in three repeats by adding a standard solution of pesticide at the levels of 0.10, 0.50, 1.0, 10, 25, 50, 100, 125, 150, 200 and 250 ppb to the sample that does not contain pesticide residue (Figure 8A,B). Average recovery values were determined and are given in Table 4. Recovery values are between 93.6–104.7% and comply with the limit values (70–120%) mentioned in the SANTE document [30]. The acquired correlation coefficient (R^2^) ranges from 0.98–0.99.

For the specification of the LOD value, blank samples were spiked with pesticides at the concentration of 0.10 ppb with ten replications. After that, the LOD value for coumaphos was determined by multiplying the blank sample (n = 10) standard deviation (s) perusals by three. The LOQ pointed out by multiplying the blank sample (n = 10) standard deviation (s) perusals by ten [31]. It can be viewed that the LOD value of coumaphos is 0.001 ppb, and the LOQ value is 0.004 ppb. The acquired correlation coefficient (R^2^) ranges from 0.98–0.99.

Recovery studies are typically conducted to evaluate the accuracy and precision of analytical methods. In this case, the recovery studies were likely conducted to determine how well the analytical techniques were able to detect coumaphos in a given sample. The recovery of an analyte in an assay is the detector response obtained from an amount of the analyte added to and extracted from the matrix. Simply recovery values were calculated using the formula: (Amount obtained/Amount added) × 100 and recorded as the percent recovery amount. All these results show that the recovery results obtained from SPR experiments were better than those obtained from LC/MS-MS experiments when testing for coumaphos. This may indicate that the SPR method is more sensitive and accurate in detecting coumaphos in the sample.

### 3.7. The Determination of Coumaphos from Honey Sample

The matrix effect was evaluated using honey samples, which were selected as a real sample to show coumaphos recognition sensitively. Honey is accepted as a healthy and comprehensive nutrient among the public. A honey sample (1:5 ratio in water) was prepared by spiking 100 ppb of coumaphos pesticide and then passed through a CIP SPR sensor system. The experimental result was obtained using a CIP SPR sensor with coumaphos spiked honey sample solution, and the ΔR value was reported as 11.17. The sensorgram for a 100 ppb coumaphos spiked honey sample was shown in Figure 9. The unspiked honey solution was also applied to evaluate the CIP SPR nanosensor signal response. So, a coumaphos spiked sample at 100 ppb concentration was applied to the CIP SPR nanosensor and detected precisely with a molecularly imprinted SPR system.

## 4. Conclusions

The development of advanced sensing systems based on molecularly imprinted polymers (MIPs) has contributed significantly to various detection methods, including electrochemical, optical, and mass-based sensors. Among them, optical sensors are becoming increasingly popular due to their high sensitivity, easy operation, and low cost. In this study, a novel CIP-SPR nanosensor was developed for the selective and sensitive detection of coumaphos, a harmful pesticide, by utilizing molecular imprinting technology. The CIP-SPR nanosensor was prepared using a template molecule, coumaphos, and advanced polymerization techniques. The shift in resonant angle was recorded during the experiments by applying different concentrations of coumaphos solution to the CIP-SPR nanosensor. Selectivity studies were performed using different competitor molecules, which are similar in size and shape to the template molecule. The results demonstrated that the prepared nanosensor was more selective to coumaphos molecules than other competitor molecules. The calculated imprinting factor (IF: 4.4) indicated that the imprinting process was performed successfully, and coumaphos was detected more selectively by the CIP-SPR nanosensor than the NIP-SPR nanosensor.

Furthermore, the adsorption mechanism was evaluated using Scatchard, Langmuir, Freundlich, and Langmuir–Freundlich adsorption models analyses. The results suggested that the interaction mechanism was compatible with the Langmuir model. The validation studies performed by LC/MSMS analyses confirmed that the results obtained by the CIP-SPR nanosensor were consistent with the LC/MS-MS results. Evaluation of the matrix effect was conducted using honey samples, which were chosen as a real sample to demonstrate the sensitivity of coumaphos recognition.

The results obtained from the various methods used to detect coumaphos present in numerous sample matrices, including honey, milk, and aqueous solutions, are summarized in Table 5.

The techniques used in the procedures include surface plasmon resonance (SPR), electrochemistry, gas chromatography, and various recognition elements such as antibodies, enzymes, and single-stranded DNA (ssDNA). The findings demonstrate that the coumaphos detection ranges highly depend on the technique and sample matrix. The EDI-OS/Antibody and ssDNA approach described by Dai et al. [33] has the lowest detection limit of 0.00018 ppb in milk samples, while the SPR/MIP method reported in this work has the lowest detection limit of 0.001 ppb in honey samples. The use of MIPs as a biorecognition element in this work can significantly reduce the cost of biosensors while providing high selectivity and sensitivity. Overall, the developed CIP SPR nanosensor allows the production of new articles describing biosensors and sensing techniques that can detect coumaphos in the future.

This study presents a simple, sensitive, and non-toxic sensing technology for the label-free determination of coumaphos, which does not require any complicated coupling processes. In conclusion, the developed CIP-SPR nanosensor using molecular imprinting technology provides a promising approach for the selective and sensitive detection of coumaphos at a low cost, which can be extended to the detection of other harmful pesticides in various environmental and agricultural applications.

## Figures and Tables

**Figure 1 micromachines-14-00707-f001:**
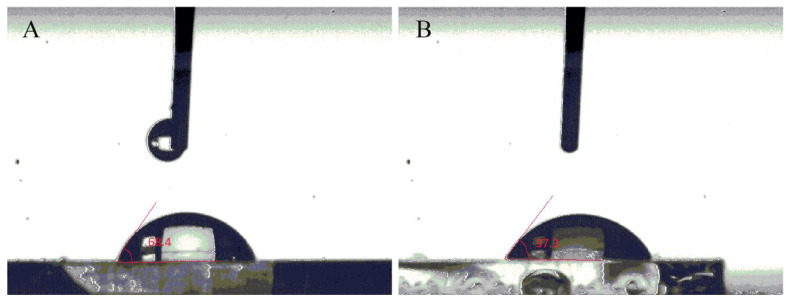
Contact angle: (**A**) CIP-SPR nanosensor, and (**B**) NIP-SPR.

**Figure 2 micromachines-14-00707-f002:**
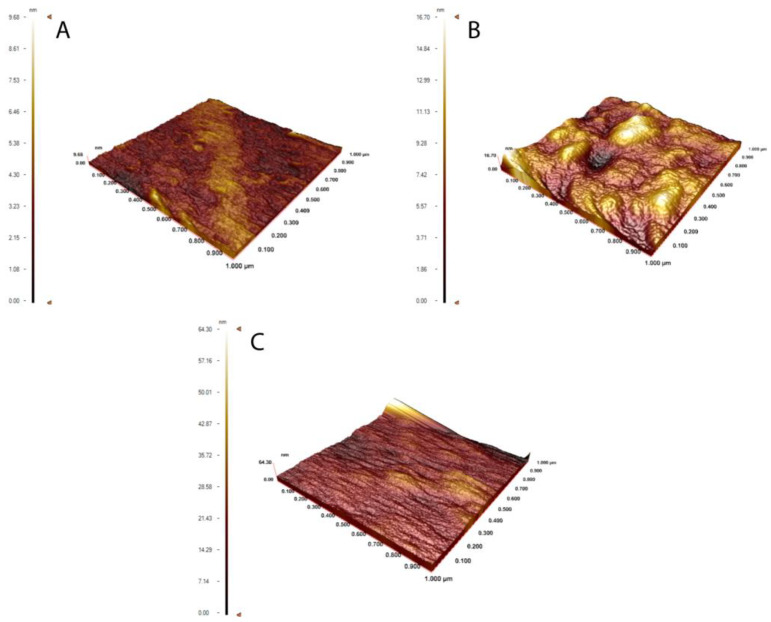
AFM Images; (**A**) bare SPR chip, (**B**) ally-modified SPR chip, and (**C**) CIP-SPR nanosensor.

**Figure 3 micromachines-14-00707-f003:**
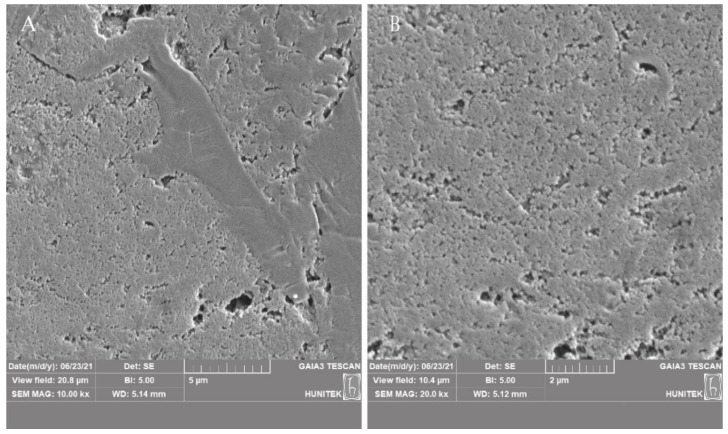
CIP-SPR nanosensor SEM images, (**A**) 10.00 kx mag, and (**B**) 20.00 kx mag.

**Figure 4 micromachines-14-00707-f004:**
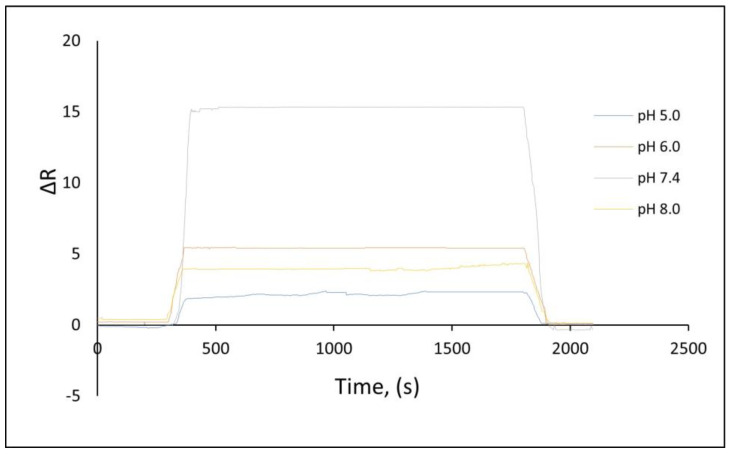
Detection sensorgram of coumaphos by designed CIP-SPR nanosensor in different pH (C: 100 ppb, t: 25 °C).

**Figure 5 micromachines-14-00707-f005:**
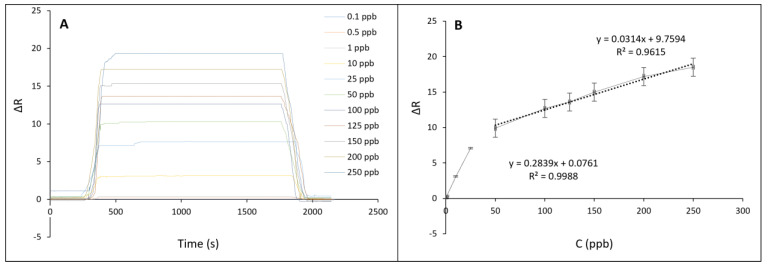
(**A**) Sensorgrams of coumaphos detection by the designed CIP-SPR nanosensor in different concentrations (pH: 7.4). (**B**) Standard calibration curve for coumaphos detection by designed CIP-SPR nanosensor in different concentrations (0.10–250 ppb, pH: 7.4).

**Figure 6 micromachines-14-00707-f006:**
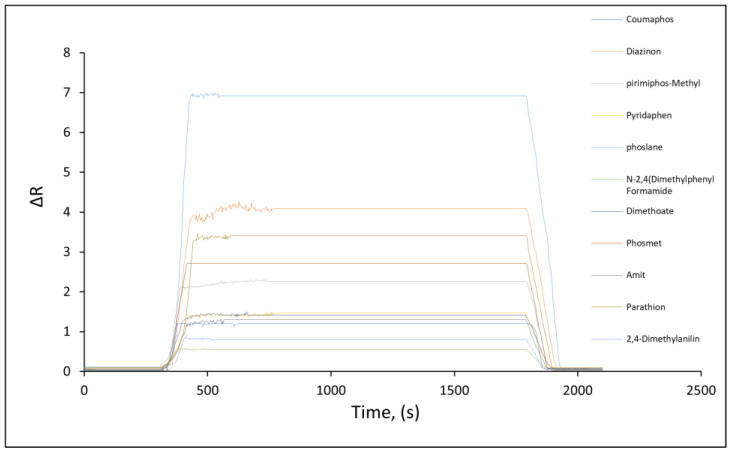
Selectivity studies of CIP-SPR nanosensor using different competitor molecules: diazinon, pirimiphos- methyl, pyridaphenthion, phosalone, *N*-(2,4-dimethylphenyl)formamide, 2,4-dimethylaniline, dimethoate, phosmet, amitraz, and parathion-ethyl (C: 50 ppb, pH 7.4, t: 25 °C).

**Figure 7 micromachines-14-00707-f007:**
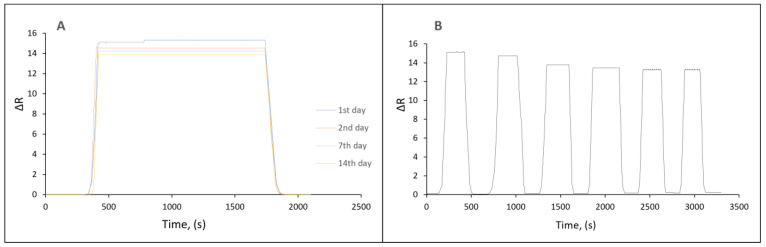
(**A**) Reusability of CIP-SPR nanosensor for coumaphos detection after fourteenth days (C: 100 ppb, pH 7.4, t: 25 °C). (**B**) Repeatability of CIP-SPR nanosensor for coumaphos detection after six measurements (C: 100 ppb, pH 7.4, t: 25 °C).

**Figure 8 micromachines-14-00707-f008:**
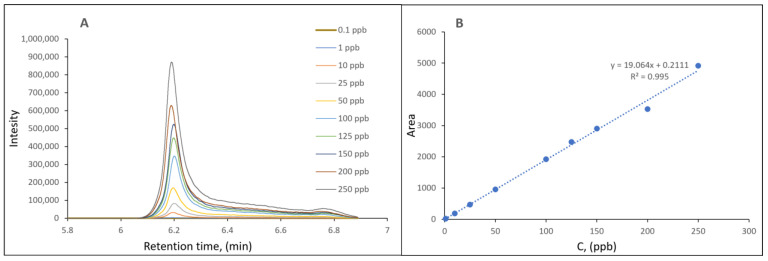
(**A**) Coumaphos LC-MS/MS retention time versus intensity graph. (**B**) Coumaphos calibration graph.

**Figure 9 micromachines-14-00707-f009:**
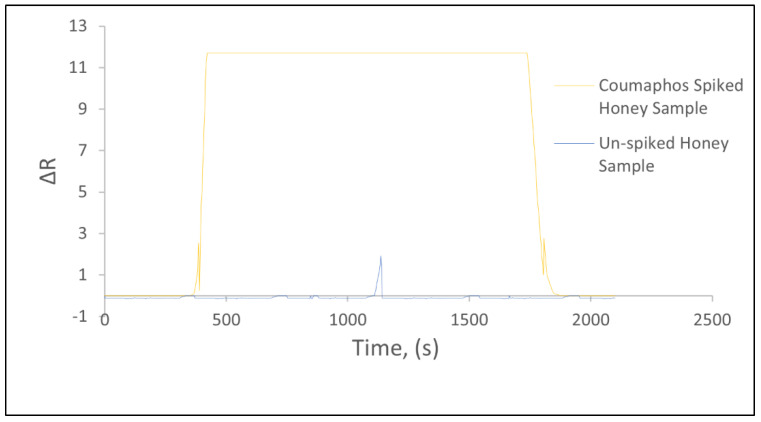
CIP SPR sensor response for Coumaphos spiked (100 ppb) and unspiked honey sample.

**Table 1 micromachines-14-00707-t001:** Equilibrium and association kinetics constants.

Association Kinetics Analysis		Equilibrium Analysis (Scatchard)	
k_a_ (L/µgs)	0.0013	ΔR_max_	23
k_d_ (1/s)	5 × 10^−4^	K_a_ (L/µg)	0.016
K_a_ (L/µg)	2.6	K_d_ (µg/L)	62
K_d_ (µg/L)	0.38	R^2^	0.91
R^2^	0.95		

**Table 2 micromachines-14-00707-t002:** The parameters of thermodynamic models isotherm.

Langmuir		Freundlich		Langmuir- Freundlich	
ΔR_max_	9	ΔR_max_	27	ΔR_max_	20
K_a_ (µg/L)	3.7	1/n	0.76	1/n	0.76
K_d_ (L/µg)	0.27	R^2^	0.97	K_a_ (L/µg)	9.35
R^2^	0.99			K_d_ (µg/L)	0.107
				R^2^	0.98

**Table 3 micromachines-14-00707-t003:** Selectivity coefficient.

	CIP-SPR		NIP-SPR		
	ΔR	k	ΔR	k	k′
Coumaphos	15		1.2		
Diazinon	2.1	7.1	4.1	0.3	23.7
Pirimiphos-methyl	3.9	3.8	2.2	0,55	6.9
Pyridaphenthion	2.8	5.4	1.3	0.92	5.9
Phosalone	3.8	3.9	6.8	0.18	21.7
*N*-(2,4-dimethylphenyl)formamide	2.8	5.3	0.55	2.1	2.5
2,4-dimethylaniline	1.8	8.3	0.8	3.7	2.2
Dimethoate	5.1	2.9	1.4	0.86	2.2
Phosmet	0.65	23.07	2.7	0.44	52.4
Amitraz	0.61	24.6	1.3	0.92	26.7
Parathion-ethyl	2.3	6.5	3.4	0.35	18.6

**Table 4 micromachines-14-00707-t004:** CIP-SPR nanosensor validation study with LC-MS/MS system.

Added Amount (ppb)	Found (ppb)		Recovery (%)	
	SPR	LC/MS-MS (Average)	SPR	LC/MS-MS
0.1	0.098	na	98	na
0.5	0.49	na	98	na
1	0.98	1.031	98	103.1
10	9.78	10.469	98	104.7
25	24.69	25.579	99	102.3
50	49.11	51.193	98	102.4
100	98.39	102.195	98	102.2
125	123.34	131.266	99	105
150	148.56	154.137	99	102.8
200	195.82	187.117	98	93.6
250	244.91	260.311	98	104.1

**Table 5 micromachines-14-00707-t005:** Studies that compare the detection of coumaphos using different methods or techniques.

Method/Recognition Element	Sample	Range	Detection Limit	Ref.
SPR/MIP	Honey	0.1–250 ppb	0.001 ppb	This work
SPR/Antibody	Aqueous	50–5000 ppb	25 ppb	[32]
EDI-OS/Antibody and ssDNA	Milk	0.0005–0.1 ppb	0.00018 ppb	[33]
Electrochemical/Enzyme	Aqueous	15–200 ppb	2 ppb	[34]
Electrochemical/Enzyme	Aqueous	6.1–183 ppb	1.5 ppb	[35]
Electrochemical/Enzyme	Honey	8–100 ppb	8 ppb	[36]
GC-FPD	Honey Propolis	0.1–1.5 ppm	12 ppb 26 ppb	[37]

## Data Availability

The authors can confirm that all relevant data are included in the article.

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
