# Peer review of "Simple and Fast Pesticide Nanosensors: Example of Surface Plasmon Resonance Coumaphos Nanosensor"

_micromachines, 2023, doi:10.3390/mi14040707_

Round 1
Reviewer 1 Report
1) This paper absorbs more audience if the authors prepare a proper graphical abstract.
2) In materials & methods, the authors prepared stock solutions from each pesticide. In lines 111-129, they reported on honey samples. It is not clear at all whether the LOD and selectivity results were obtained from pure pesticides or from honey samples. Also, how do they know their biosensor works on real samples?
3) It seems their biosensor works on various pesticides. Usually, a biosensor would be designed against a specific molecule/target and does not interact with other molecules. (specificity) However, specificity does not mention here.
4) This manuscript does not have discussion at all. In the discussion, the authors must compare their results with previous research. It would be nice to summarize the discussion part is a table (current study at the top of the table and relevant results) in both sensitivity and specificity. The lack of this comparison makes it impossible to publish in micromachines.
Author Response
Reviewer 1
Comments and Suggestions for Authors
- This paper absorbs more audience if the authors prepare a proper graphical abstract.
Thank you for your valuable recommendation. We included graphical abstract as you suggested.
- In materials & methods, the authors prepared stock solutions from each pesticide. In lines 111-129, they reported on honey samples. It is not clear at all whether the LOD and selectivity results were obtained from pure pesticides or from honey samples. Also, how do they know their biosensor works on real samples?
Thank you for your careful evaluation. To make clarify real sample analyses, we added explanations into the section of “2.4. Applying CIP-SPR Nanosensor for Detection of Coumaphos and Kinetic Studies with Designed CIP-SPR Nanosensor” as follow;
In addition, analyzes were carried out with real samples to investigate the effects of existing residues. Because the presence of the residues in the analyzed matrix effect the signal response. All honey stocks were diluted with deionized water in a 1:5 ratio. The matrices used in this study were honey collected from organically grown farms purchased from a local store.
Also we added the related explanations into the Result and discussion section as a title 3.7 The determination of Coumaphos from milk sample.
The matrix effect was evaluated using honey samples, which were selected as a real sample to show coumaphos recognition sensitively. Honey is accepted as a healthy and comprehensive nutrient among the public. A honey sample (1:5 ratio in water) was prepared by spiking 100 ppb of coumaphos pesticide and then passed through a CIP SPR sensor system. The experimental result was obtained using a CIP SPR sensor with coumaphos spiked honey sample solution, and the ΔR value was reported as 11.17. The sensorgram for 100 ppb coumaphos spiked honey sample was shown in Figure 9. The un-spiked honey solution was also applied to evaluate the CIP SPR nanosensor signal response. So coumaphos spiked sample at 100 ppb concentration was applied to the CIP SPR nanosensor and detected precisely with molecularly imprinted SPR system.
3) It seems their biosensor works on various pesticides. Usually, a biosensor would be designed against a specific molecule/target and does not interact with other molecules. (specificity) However, specificity does not mention here.
Thank you for your careful evaluation. In this study, SPR-based nanosensor was prepared for the selective and sensitive detection of organophosphate-based coumaphos pesticide. The prepared biosensor works on various pesticides but as mentioned in the section of “3.4. Selectivity Studies” the selectivity coefficients (k) for competitor molecules (diazinon, pirimiphos- methyl, pyridaphenthion, phosalone, N-(2,4-dimethylphenyl)formamide, 2,4-dimethylaniline, dimethoate, phosmet, amitraz, and parathion- ethyl respect to coumaphos were calculated using CIP-SPR nanosensor, individually. As seen in Table 3, the selectivity constants (k= ΔRcoumaphos/ΔRcompetitor), for the coumaphos pesticide is higher than the competitor molecules. For example CIP-SPR nanosensor 7.1 times more selective for coumaphos than competitor Diazinon molecule while 24.6 times more selective than Amitraz pesticide. The imprinting efficiency of CIP SPR nanosensor for Coumaphos detection was evaluated by comparing it with the non-imprinted NIP SPR sensor to indicate the specifity of the prepared CIP SPR nanosensor for Coumaphos detection. The high value of the calculated imprinting factor (I.F.4.4) proved that the CIP SPR nanosensor detects coumaphos pesticide specifically. We incorporated explanations with red words into the section as follow;
As seen in Table 3, CIP-SPR nanosensor 7.1, 3.8, 5.4, 3.9, 5.3, 8.3, 2.9, 23.07, 24.6, 6.5 times more selective for the template coumaphos pesticide than competitor diazinon, pirimiphos-methyl pyridaphenthion, phosalone, N-(2,4-dimethylphenyl)formamide, 2,4-dimethylaniline, dimethoate, phosmet, amitraz and parathion-ethyl molecules, respectively. The high ratio of selectivity values indicates that CIP-SPR nanosensor was prepared successfully for selective detection of coumaphos pesticide. The imprinting efficiency of CIP SPR nanosensor for coumaphos detection was evaluated by comparing it with the non-imprinted NIP SPR nanosensor to indicate the specifity of the prepared CIP SPR nanosensor for Coumaphos detection. The high value of the calculated imprinting factor (I.F.4.4) proved that the CIP SPR nanosensor detects coumaphos pesticide specifically. Additionally, linear relationship for the 0.1–250 ppb coumaphos concentration range reaching as low as low limits of detection and limit of quantification (LOQ: 0.001 and LOQ: 0.003 ppb) values indicates that the MIP combined with SPR method was used successfully for the specific determination of comaphos pesticide.
CIP-SPR |
NIP-SPR |
|
|||
ΔR |
k |
ΔR |
k |
kˊ |
|
Coumaphos |
15 |
|
1.2 |
|
|
Diazinon |
2.1 |
7.1 |
4.1 |
0.3 |
23.7 |
Pirimiphos-methyl |
3.9 |
3.8 |
2.2 |
0,55 |
6.9 |
Pyridaphenthion |
2.8 |
5.4 |
1.3 |
0.92 |
5.9 |
Phosalone |
3.8 |
3.9 |
6.8 |
0.18 |
21.7 |
N-(2,4-dimethylphenyl)formamide |
2.8 |
5.3 |
0.55 |
2.1 |
2.5 |
2,4-dimethylaniline |
1.8 |
8.3 |
0.8 |
3.7 |
2.2 |
Dimethoate |
5.1 |
2.9 |
1.4 |
0.86 |
2.2 |
Phosmet |
0.65 |
23.07 |
2.7 |
0.44 |
52.4 |
Amitraz |
0.61 |
24.6 |
1.3 |
0.92 |
26.7 |
Parathion-ethyl |
2.3 |
6.5 |
3.4 |
0.35 |
18.6 |
4) This manuscript does not have discussion at all. In the discussion, the authors must compare their results with previous research. It would be nice to summarize the discussion part is a table (current study at the top of the table and relevant results) in both sensitivity and specificity. The lack of this comparison makes it impossible to publish in micromachines.
Thank you for your careful evaluation. As you suggested we discussed and compared the obtained results in the results and discussion in section 3.8.
3.8 Related research for coumaphos detection
The results obtained from the various methods used to detect coumaphos present in numerous sample matrices, including honey, milk, and aqueous solutions, are summa-rized in Table 5. The techniques used in the procedures include surface plasmon reso-nance (SPR), electrochemistry, gas chromatography, and various recognition elements like antibodies, enzymes, and single-stranded DNA (ssDNA). The findings demonstrate that the coumaphos detection ranges are highly depending on the technique and sample ma-trix. The EDI-OS/Antibody and ssDNA approach described by Dai et al. [33] has the low-est detection limit of 0.00018 ppb in milk samples, while the SPR/MIP method reported in this work has the lowest detection limit of 0.001 ppb in honey samples. The use of MIPs as a biorecognition element in this work can significantly reduce the cost of biosensors while providing high selectivity and sensitivity. Overall, the developed CIP SPR nanonosensor allows the production of new articles describing biosensors and sensing techniques that can detect coumaphos in the future.
Table 5. Studies that compare the detection of coumaphos using different methods or techniques.
Method/Recognition Element |
Sample |
Range |
Detection Limit |
Ref |
SPR/MIP |
Honey |
0.1-250 ppb |
0.001 ppb |
This work |
SPR/Antibody |
Aqueous |
50-5000 ppb |
25 ppb |
[30] |
EDI-OS/Antibody and ssDNA |
Milk |
0.0005-0.1 ppb |
0.00018 ppb |
[31] |
Electrochemical/Enzyme |
Aqueous |
15-200 ppb |
2 ppb |
[32] |
Electrochemical/Enzyme |
Aqueous |
6.1-183 ppb |
1.5 ppb |
[33] |
Electrochemical/Enzyme |
Honey |
8-100 ppb |
8 ppb |
[34] |
GC-FPD |
Honey Propolis |
0.1-1.5 ppm |
12 ppb 26 ppb |
[35] |

Reviewer 2 Report
Th authors presented a surface plasmon resonance-based nanosensor for the pesticide coumaphos. It is of interest to multiple fields. However, the authors need to make sure the data they presented are consistent, especially the Ka and Kd values in text and one of the tables. There are other minor revisions to be made and minor problems to be addressed. Please see the attached comments.

Author Response
Reviewer 2
Th authors presented a surface plasmon resonance-based nanosensor for the pesticide coumaphos. It is of interest to multiple fields. However, the authors need to make sure the data they presented are consistent, especially the Ka and Kd values in text and one of the tables. There are other minor revisions to be made and minor problems to be addressed. Please see the attached comments.a
Thank you for your careful evaluation. As you suggested we made uniform and consistent the data that we presented.
Line |
Original |
Comments |
response |
61-62 |
[US Environmental Protection Agency (EPA) 1998, Extremely 61 Hazardous Substances (EHS), Chemical Profiles and Emergency First Aid Guides. Wash-62 ington, DC, US, Government Printing Office] |
Shouldn’t this be in the reference section? |
Reference inserted into the related section as; https://www.ecfr.gov/current/title-40/chapter-I/subchapter-J/part-355 |
73-75 |
Also, it's noted that the transportation time between food production and terrible transportation conditions could contaminate quality control laboratory foods, weather, fungi, and other microorganisms, which may cause unreliable analysis results [21]. |
This sentence needs to be rewritten. ” …could contaminate quality control laboratory foods, weather, fungi, and other microorganisms”? |
The sentence was rewritten as; Also, it's noted that the elapsed time between food production and the shipping conditions can cause contamination of the food with fungi and other microorganisms, resulting in unreliable analysis results of the quality control laboratory. |
304-305 |
Table 1 shows the association rate constant (ka: 0.0013 L/μ gs) and disassociation constant (kd: 9×10-5 1/s)
|
disassociation rate constant |
We changed as; dissociation rate constant |
307-308 |
In addition, the equilibrium association constant (Ka: 1.4 L/μ g) and the disassociation constant (Kd: 0.7 μg/L)
|
In table 1 , Ka is 14.3 L/g
Kd in Table 1 is 0.07 g/L, not 0.7.
Please double check. |
We corrected the wrong written numbers in Table 1 and made uniform throughout manuscript. |
345-348 |
|
Needs to be rewritten to make it more clear. Pay attention to parenthesis. |
We have rewritten the sentence as; The selectivity coefficients (k) and relative selectivity coefficients (kʼ) for coumaphos pesticide were recorded individually by applying the competitor molecules (diazinon, pirimiphos-methyl, pyridaphenthion, phosalone, N-(2,4-dimethylphenyl)formamide, 2,4-dimethylaniline, dimethoate, phosmet, amitraz, and parathion-ethyl) to the CIP-SPR and NIP-SPR nanosensors. |
376 |
siting binds
|
Binding sites? |
We corrected as; Binding sites
|
389 |
were spiked pesticides
|
were spiked with pesticides
|
We corrected as; were spiked with pesticides
|
398-401 |
All these results shows that the recovery results obtained from SPR experiments were better than the recovery results obtained from LC/MS-MS experiments when testing for coumaphos.
|
How were the recovery rates from SPR determined? |
Determination of recovery rates was explained in the section”3.6. Validation studies” as follow; The recovery of an analyte in an assay is the detector response obtained from an amount of the analyte added to and extracted from the matrix. Simply recovery values were calcu-lated using formula: Amount obtained/Amount added*100 and recorded as pecent recov-ery amount. |

Round 2
Reviewer 1 Report
The authors answered my comments. However, Two issues still are remained:
1) I have not found the graphical abstract.
2) This is academic writing. Some short writing is not appropriate. e. g line 76. "it's" is not formal in academic text.
Author Response
Reviewer 1;
Comments and Suggestions for Authors
The authors answered my comments. However, Two issues still are remained:
1) I have not found the graphical abstract.
Thank you for your careful evaluation and recommendation. We inserted the graphical abstract formerly, and we are now uploading the graphical abstract again as follows ;
2) This is academic writing. Some short writing is not appropriate. e. g line 76. "it's" is not formal in academic text.
We checked the manuscript thoroughly, and we made corrections to the manuscript with red words (including line 76. "it's")

Reviewer 2 Report
Line 201: Change "analyzes" to "analyses".
There are significant changes to the data listed in Table 1, not just simple changes to decimal points. This reviewer initially thought the data inconsistency in Table 1 and its associated text was due to the misplacement of decimal points. However, in the revised version, the values in the first column in Table 1 are totally different from those in the original version (see the attached file). Why are there such changes? How can one assure the reliability of such data?

Author Response
- Line 201: Change "analyzes" to "analyses"
Line 201 was changed.
- There are significant changes to the data listed in Table 1, not just simple changes to decimal points. This reviewer initially thought the data inconsistency in Table 1 and its associated text was due to the misplacement of decimal points. However, in the revised version, the values in the first column in Table 1 are totally different from those in the original version (see the attached file). Why are there such changes? How can one assure the reliability of such data?
Thank you for your careful evaluation and suggestions. As you mentioned, in the first revised version, the values in the first column in Table 1 differ from those in the original version. Because in the first manuscript, you noticed the unit inconsistency and asked us to enable the uniformity of units in the revised manuscript. While controlling the units, we recalculated all the values present in the table. And we also noticed that some values were miscalculated. Again, we calculated the values, and the last version we supplied is correct, as in the table below.
Table 1. Equilibrium and association kinetics constants.
Association kinetics analysis |
Equilibrium analysis (Scatchard) |
||
ka (L/µgs) |
0.0013 |
ΔRmax |
23 |
kd (1/s) |
5×10-4 |
Ka (L/µg) |
0.016 |
Ka (L/µg) |
2.6 |
Kd (µg/L) |
62 |
Kd (µg/L) |
0.38 |
R2 |
0.91 |
R2 |
0.95 |
|
|
Table 1 shows the association rate constant (ka: 0.0013 L/µgs) and dissociation rate constant (kd: 5×10-4 1/s) of interaction between coumaphos and CIP-SPR nanosensor demonstrate a tendency to association between the coumaphos molecules and CIP-SPR nanosensor. In addition, the equilibrium association constant (Ka: 2.6 L/µg) and the dissociation constant (Kd: 0.38 µg/L) represent the high binding affinity between the analyte and nanosensor.

Round 3
Reviewer 2 Report
Since the author responded that the data in the last revision was accurate, my concern has been addressed.
In the future, please alert the reviewer when material changes to data are made.